# Institutional distance, trade agreements, and intellectual property trade networks: Evidence from cross-border data

Yida Wang[1], Jiangjiao Wang[2]*

**1** School of International Trade and Economics, Anhui University of Finance and Economics, Bengbu, Anhui, China, **2** Jiyang College of Zhejiang A&F University, Zhuji, Zhejiang, China

* 673826599@qq.com

**Data Availability Statement:** All relevant data can be found at the following location: DOI: 10.57760/sciencedb.09553.

**Funding:** This study was supported in the form of funding by the General Scientific Research Projects

## Abstract

Based on the Temporal Exponential Random Graph Models (TERGM), this paper applies global intellectual property trade data between countries to investigate the impact and mechanism of institutional distance on the intellectual property trade network. The study finds that the smaller the institutional distance between countries, the more conducive it is to build an intellectual property trade network. This conclusion remains valid after controlling for geographical adjacency, use of a common language, existence of colonial relationships, and characteristics of the intellectual property trade network. Moreover, through regression by year, it is found that this impact increases year by year. Further, after regressing on sub-indicators of institutional distance, it is found that the smaller the distance in political stability, government efficiency, and regulatory quality, the greater the probability of generating an intellectual property trade relationship. Mechanism analysis reveals that economies with smaller institutional distances are more likely to sign trade agreements, thereby generating trade relationships and promoting the establishment of intellectual property trade networks. In order to deeply participate in the intellectual property trade network, countries should actively align with international institutional norms and sign bilateral or multilateral trade agreements with countries with similar institutional levels to enhance the production level and export of intellectual property.

## Introduction

With the advent of the knowledge economy era, intellectual property trade based on cross-border transfer or licensing of intellectual property rights gradually became one of the three major trade in the world. As of 2017, intangible capital such as intellectual property created 30.4% of the value of global manufactured goods trade, determining the success rate of products in the market and serving as a strategic resource for the country to improve its core competitiveness. The creation of intellectual property has evolved from being monopolized by highly developed countries in the past to moving towards a global value chain division of labor. In recent years, the global economy is in a prolonged state of stagnation, leading to escalating political and

of Zhejiang Education Department (Y202353320) and the Shaoxing Higher Education Teaching Reform Project (SXSJG202305).

**Competing interests:** The authors have declared that no competing interests exist.

economic tensions among nations, resulting in a growing frequency of trade frictions related to intellectual property.

TRIPs and the WTO have had a huge impact as multilateral trade dispute resolution mechanisms, but in the new situation, the traditional TRIPs and WTO-centered multilateral negotiations gradually lose their impact, while the various institutional requirements for the development of intellectual property trade become increasingly stringent and complex. This trend pushes the evolution of the intellectual property network toward two extremes: on one hand, developed countries seek to maintain their dominance in the global trade value chain by continuously raising the standards of intellectual property protection, which actively advocate for the establishment of multilateral, bilateral, and regional international agreements, aiming to elevate their national policies to global standards, thereby expanding their intellectual property policies to maximize their national interests on a global scale. On the other hand, developing countries and certain civil society forces are committed to building a more equitable framework for intellectual property protection. They aim to break away from past global trade agreements with high levels of intellectual property protection and, instead, seek bilateral or multilateral trade agreements while formulating policies that are more inclusive and sustainable, all with the objective of ensuring the stability and fairness of intellectual property trade. Among the many factors that affect the construction of bilateral intellectual property trade networks, institutional differences are receiving increasing attention. Institutional distance may result in multinational enterprises facing varying intellectual property environments in different countries, increasing compliance costs and legal risks. This not only significantly impacts a firm's global strategy and competitiveness but also underscores the international community's growing emphasis on global coordination and consistency in intellectual property rights.

Institutional distance primarily refers to the differences in formal institutions among countries, including legal systems, political stability, government efficiency, regulatory quality, and other aspects. Good institutions can effectively allocate resources and reduce transaction costs, while poorer institutional environments lack protection for transactional rationality, thus dampening trading enthusiasm. The intellectual property trade network is a multilateral trade network centered around intellectual property rights, primarily encompassing activities such as licensing and transfer of intellectual property. These transactions are characterized by heterogeneity, intangibility, and complexity, making the influence of institutional environments more direct and crucial. Countries with high-level institutional environments generally exhibit higher intellectual property output. However, when engaging in trade with countries with low intellectual property output but also low-level institutional environments, the transactional risks are heightened. The relationship between the two has sparked controversy. Thus, does institutional distance influence the network of intellectual property trade? If there is an influence, what is the mechanism of this impact? Studying the relationship between institutional distance and the network of intellectual property trade, as outlined in the previous question, can provide insights and policy implications for reconciling global intellectual property conflicts, advancing institutional reforms in various countries, and reducing risks in international intellectual property trade.

At present, there is limited academic research on the impact of institutional distance on the intellectual property trade network, but there is a more extensive discussion on both separately. In the terms of intellectual property trade, with the development of the intellectual economy, the intellectual property market has become a crucial foundation for a country to promote innovation and enhance its national trade competitiveness [1]. The maturity of a country's market institutions significantly affects the willingness of host countries to import their intellectual property [2]. Due to the contractual nature of intellectual property trade, institutional elements inevitably become the focus of research, among which the level of

intellectual property protection is widely scrutinized by domestic and international scholars. Enhancing the level of intellectual property protection can effectively raise a country's innovation level and improve the structure of intellectual property trade [3]. In terms of institutional distance, since first introduced the concept of institutional distance, numerous scholars have engaged in discussions on the relationship between institutional distance and international trade [4]; Based on the New-New Trade Theory and New Institutional Economics, overall institutional quality can positively impact international trade through means such as transaction costs and incomplete contracts [5]. From an economic institutional perspective, the average institutional distance constructed can affect foreign trade through the suppression of transaction costs [6]. From the viewpoint of New Institutional Economics, formal and informal institutional distances have different effects on service trade [7]; From the perspective of multinational enterprises, institutional distance significantly affects the performance of overseas subsidiaries [8]. In the context of bidirectional talent flow in technology between countries, institutional distance plays a significant role [9]. Different levels of institutional distance have varied impacts on economic activities; countries' outward investments exhibit characteristics of "political institutional proximity" and "economic institutional escape" [10]. Another portion of the literature provides evidence of the role of trade agreements in promoting intellectual property trade. Trade agreements that include levels of intellectual property protection can facilitate inter-country intellectual property trade and promote the upgrading of their industrial chains [11]; Regional trade agreements can enhance the facilitating role of institutional quality in intellectual property trade [12]; Regional trade agreements can help bridge institutional gaps and mitigate the impact of bilateral institutional distance [13]. For developing countries, the intellectual property clauses in trade agreements may lead to short-term losses due to high patent fees and lower returns [14]. There is also an analysis based on mega trade agreements such as RCEP, which offers recommendations for countries participating in the formulation of rules for intellectual property trade [15]. Regional trade agreements can enhance the facilitating role of institutional quality in intellectual property trade [12]. In terms of research methods, most scholars examined the probability of institutional distance and trade relationships based on traditional trade gravity models and spatial models, including discussions from various perspectives such as adjacent effects and national heterogeneity, the impact intensity of various institutions [16], controlling for culture and geographic distance [17]. However, as the intellectual property rights trade network is a highly important part of international trade networks, with the development of technology and information technology, traditional spatial and gravity models may find it difficult to detect the internal interdependencies and embedded external relationships. Overall, existing literature on the effects of institutional distance on various types of international economic activities is abundant, demonstrating the significant impact of institutional distance in international trade research. However, it still fails to validate the specific effects of institutional distance on intellectual property trade. Moreover, research on intellectual property trade networks is limited, and most existing literature relies on traditional trade gravity models, making it difficult to explore the network characteristics of intellectual property trade. Consequently, it is challenging to explain the forms of intellectual property trade between countries with different institutional distances in reality. Furthermore, in the international intellectual property trade landscape reshaped by multilateral trade agreements, the aforementioned studies still offer limited insights into understanding institutional distance, trade agreements, and intellectual property trade networks.

Therefore, this paper, based on the temporal exponential random graph model (TERGM), incorporates institutional distance and the intellectual property trade network into the research framework. Furthermore, it validates the model's conclusions using the generalized

exponential random graph model (GERGM) to explore the mechanisms and characteristics of the dynamic development of the network. The innovation of this paper lies in the following aspects: (1) In terms of the research object, this paper differs from traditional studies of structural characteristics of bilateral trade relationships. Instead, it examines the formation of dyads and triads within the trade network as the object of study to explore the dynamic mechanisms of its evolution. This approach aims to deconstruct the embedded patterns of institutional distance matrix that influence the intellectual property trade network. (2) In terms of the research perspective, this paper takes a global perspective on the intellectual property trade network, controlling for potential influencing factors from both importing and exporting countries. It considers the roles of external scenarios and internal dependencies, resulting in more comprehensive and robust conclusions. (3) In terms of research methodology, this paper employs the exponential random graph model, which allows for the simultaneous control of various influencing factors and mechanisms, such as node attributes, internal, and external mechanisms. It also enables simulation based on model parameters, thereby providing a reference for the adjustment and improvement of the structure of the intellectual property trade network.(4) In terms of research framework, this article introduces a dynamic research framework that encompasses power relations, geopolitical strategies, and economic interests. It delves into the complexity inherent within institutional distances, intellectual property trade networks, and trade agreements from a deeper socio-political dimension.

## Theoretical analysis and research hypotheses

Intellectual property trade, as an essential aspect of services trade, inherits its characteristics of heterogeneity, intangibility, and complexity, resulting in networked features. In terms of the impact of institutional distance on intellectual property trade relationships: firstly, from the perspective of transaction costs, the existence of institutional distance between countries increases the ex-ante and ex-post costs of intellectual property trade, such as judicial risks, negotiation costs, information costs, etc. [16, 18, 19]. When institutional distance is significant, the exporting country faces severe information asymmetry [20] and higher market entry barriers in the importing country [5]. Excessive transaction costs can reduce the willingness of the exporting country to engage in intellectual property trade. Secondly, from the perspective of contractual risk, the essence of a transaction is the transfer of property rights, and institutions restrict the potential opportunistic behaviors in defining, protecting, and trading these property rights during this process [21]. Intellectual property trade belongs to a trade type with high contract intensity, and exporting countries tend to have relatively more exports in such industries. They have a weaker motivation to form trade relationships with countries with weak institutional environments [18, 22]. Additionally, due to the limited rationality of the parties involved, there may be a risk of 'hold-up' in the transaction, and due to incomplete institutions, they may face commitments being violated in areas such as taxation and property rights protection [5]. Therefore, countries exporting intellectual property tend to seek trade partners with institutional levels similar to their own. In terms of the impact of institutional distance on the formation of the intellectual property trade network: from the perspective of third-party effects, when the two countries involved in intellectual property trade have a significant institutional distance from other countries, the trade costs and risks for third countries are higher, resulting in a noticeable trade diversion effect [23]. This effect leads to trade creation in higher-level countries [24]. From the perspective of value chain specialization, several countries with similar institutional levels, by having partially integrated trade networks with closer trade governance and lower information costs [19, 25], and with knowledge products in which the additional trade costs generated by contractual and intellectual property systems are

similar across countries [26], gradually develop a trend of specialization [27], thereby deepening the intellectual property trade network. H1 has been validated.

H1: Institutional distance has a negative impact on the intellectual property trade network.

As the multilateral trade negotiations centered around the WTO stall and the diminishing influence of TRIPS, WTO member countries are turning to bilateral or multilateral cooperation. This shift has given rise to a plethora of flexible and open regional trade agreements, and these trade pacts, freely entered into by two countries, place increasing demands on the alignment of institutional systems between nations [28]. On one hand, a smaller institutional distance implies closer levels of supervision and a higher degree of rule perfection. The efficiency of legislative, administrative, and dispute resolution institutions in trade agreements is influenced by the quality of the contracting countries' institutions [29]. This allows countries with smaller institutional distance to have smoother negotiations in comprehensive agreements involving intellectual property, environmental regulations, and more. Additionally, the smaller the institutional distance, the higher the cultural identity between two countries [12], which reduces the impact of regulatory trade barriers from the other country. Countries with smaller heterogeneity have more flexibility in negotiations and are more aligned in their goals of increasing welfare levels [29]. Therefore, they are more likely to enter into trade agreements. On the other hand, countries that sign trade agreements gain advantages in many aspects necessary for constructing intellectual property trade networks. Firstly, trade agreements can promote the formation of intellectual property trade relationships by refining the international division of knowledge production and reducing the transaction costs of trade. Trade agreements not only substantially reduce tariffs but also save information gathering costs. The reduction in transaction costs greatly amplifies intermediate goods trade [30, 31], thus promoting the refinement of international specialization. Secondly, trade agreements increase the probability of engaging in innovation cooperation. Lin [32] found that service trade agreements containing intellectual property clauses can promote a country's production and export of intellectual property through technology spillover effects. Heterogeneity analysis indicates that this effect is more pronounced between countries with smaller institutional distance. Thirdly, the signing of trade agreements can reduce disputes and friction in intellectual property trade [33]. According to research by Feng [34], trade agreements signed by two countries with a high degree of trade dependence significantly reduce the expected benefits and increase the expected costs of initiating trade disputes, thereby reducing the likelihood of trade disputes. This effect is particularly evident in intellectual property trade, where disputes are more likely to occur. Last but not least, more sophisticated regulatory agencies and dispute resolution mechanisms greatly reduce the risk of intellectual property trade infringement and increase the probability of successful bilateral trade negotiations. Therefore, this paper posits that trade agreements act as intermediary variables influencing the relationship between institutional distance and the formation of intellectual property trade, and puts forward hypothesis 2.

H2: The smaller the institutional distance between countries, the greater the likelihood of reaching trade agreements, thereby promoting the formation of intellectual property trade networks.

## Model design and data sources

### Model construction

The Exponential Random Graph Model (ERGM) is a quantitative model used by domestic and international scholars to analyze emerging relational data in social networks. It is

considered one of the most effective empirical tools in the field of social network science. This model primarily examines the dependency of relationships within networks. Additionally, it can consider various hierarchical network structural variables to explore network structure and its formation processes. It achieves this by controlling for locally generated network characteristics through systematic simulation and modeling [35].

Therefore, this paper employs the Exponential Random Graph Model (ERGM) to analyze the self-organizing characteristics of the international trade network, node attribute structures, and external environmental factors. Building upon this analysis, it introduces a time variable and utilizes the resulting model, the Temporal Exponential Random Graph Model (TERGM), as the primary research framework. Furthermore, this study conducts annual regressions using the Generalized Exponential Random Graph Model (GERGM) to address potential data loss issues that may arise from binarizing the intellectual property trade network. To overcome potential efficiency and variability concerns associated with the Markov Chain Monte Carlo Maximum Likelihood Estimation (MCMC-ML) method [36], this paper employs a more efficient and robust pseudo-likelihood estimation approach.

## Variable selection

**Dependent variable: Intellectual property trade network.**   This paper's core dependent variable is the matrix of intellectual property trade volume. The matrix is structured as a directed weighted network with rows representing exporting countries and columns representing importing countries. The TERGM model does not allow weighted target networks, so when considering the temporal dimension, it is necessary to transform the international trade relations network into a standard binary network. Following the approach of Wang [37], this paper uses a threshold of one hundred thousand US dollars to convert the weighted network into a binary network. If country $i$'s annual export to country $j$ is greater than or equal to one hundred thousand US dollars, then $e_{ij} = 1$, otherwise $e_{ij} = 0$.

**Explanatory variables.**   The main explanatory variable in this study is institutional distance. We use the Worldwide Governance Indicators (WGI) provided by the World Bank as the source of this variable. The WGI includes six sub-indicators: Voice and Accountability (VA), Political Stability and Absence of Violence/Terrorism (PS), Government Effectiveness (GE), Regulatory Quality (RQ), Rule of Law (RL), and Control of Corruption (CC). We calculate the institutional distance using the distance calculation method proposed by Kogut and Singh (1988) [33]:

$$dis\_ins_{ij} = \frac{1}{6} \sum_{n=1}^{6} \frac{\left(I_{ni} - I_{nj}\right)^2}{V_n} \tag{1}$$

Here, $n$ represents the various sub-indicators of the Worldwide Governance Indicators, $I$ denotes the scores of country $i$ and $j$ on each sub-indicator, and $V$ represents the variance of institutional scores across sub-indicator $n$. The introduction of $V$ helps address comparability issues that may arise between different indicators. It is commonly believed that institutional distance is highly correlated with the economic development levels of countries. This feature could lead to endogeneity issues in the model. To mitigate this effect, following the approach by Wang [38], we orthogonalize institutional distance with respect to GDP.

## Endogenous variables

In terms of endogenous variables, this paper introduces mutual, ttriple, transitiveties, and cyclicalties, as well as out-2-stars and in-2-stars, and the outdegree and indegree of the

**Table 1. Endogenous variables.**

| Variable Name | Statistic | Statistical Significance |
|---|---|---|
| Edges | $\sum_{i,j} x_{ij}$ | Similar to the constant term in linear models, it is typically not explained |
| Mutual | $\sum_{i<j} x_{ij} x_{ij}$ | Whether there is a reciprocal relationship between two economies in the network |
| Ttriple | $\sum_{\exists k, x_{ij} x_{jk} x_{ik}=1} x_{ij} x_{jk} x_{ik}$ | Calculate the number of edges embedded in triadic closure relationships in the network. |
| Transitiveties | $\sum_{i,j,k} x_{ji} x_{ki} x_{jk}$ | The tendency of the following trade relationships between the three economies $(i, j, k)$: $i$ exports to $j$, $j$ exports to $k$, and $i$ re-exports to $k$. |
| Cyclicalties | $\sum_{i,j,k} x_{ji} x_{jk} x_{ki}$ | The tendency of the following trade relationships between the three economies $(i, j, k)$: $i$ exports to $j$, $j$ exports to $k$, and $k$ re-exports to $i$. |
| Out-2-stars | $O = \sum_{i} C^2_{\sum_j x_{ij}}$ | The trend of exporting by economic entities. |
| In-2-stars | $I = \sum_{i} C^2_{\sum_j x_{ij}}$ | The export trends of other economies to this country |
| Outdegree | $\sum_{k=2}^{N-1} (-1)^k \frac{O}{\lambda^{k-2}}$ | The activity level of an economic entity |
| Indegree | $\sum_{k=2}^{N-1} (-1)^k \frac{I}{\lambda^{k-2}}$ | The popularity of an economic entity |

exporting country as control variables. Mutual reflects the impact of 'reciprocal' relationships on the formation of connections, meaning that one-way trade relationships are more likely to develop into bilateral trade relationships; ttriple measures the number of triangles in triadic relations, while transitiveties and cyclicalties gauge the phenomenon of trade clustering in the intellectual property trade network. Out-2-stars and in-2-stars reflect the level of activity and popularity of the exporting country. The forms of each endogenous mechanism variable are as the Table 1.

**Exogenous variables.** To examine the influence of historical, cultural, and geopolitical factors in the intellectual property trade network, this study incorporates three types of exogenous relational network covariates: border network, language network, and colonial network. These correspond to the geopolitical relationships, linguistic connections, and colonial historical relationships among trading entities, to investigate the impact of the 'embedded' networks on the intellectual property trade network.

**Node attribute variables.** In terms of node attribute variables, this study selects service trade exports, merchandise trade exports, Gross Domestic Product (GDP), and foreign direct investment as proxies to control for the influence of a country's own attributes on the formation of intellectual property trade relationships. The natural logarithm of GDP is taken to address right-skewed data. Furthermore, from the perspective of the structural characteristics of node attributes [35], the study incorporates the sending effect and receiving effect of control variables into the model, corresponding to the attributes of the exporting and importing countries, respectively.

## Data sources

The data for this study primarily come from the following sources: First, import and export data are sourced from the World Trade Organization-Organization for Economic Co-operation and Development Services Trade Balance Dataset (BaTIS). After threshold conversion, a binary matrix of intellectual property trade for 143×142 countries/regions is obtained. Second,

institutional distance is derived from the World Bank's Global Governance Indicators, calculated based on six sub-indicators to generate a 143×142 matrix of institutional distance. Third, exogenous relationship networks, including adjacency networks, language networks, and colonial networks, are compiled from the CEPII database. Fourth, the bilateral trade agreement network comes from the Design of Trade Agreements (DESTA) database at the University of Bern. Fifth, other control variables are sourced from publicly available World Bank data. To mitigate the influence of the 2008 financial crisis and the 2019 global public health emergency on this study, the final dataset comprises intellectual property trade network data for 143×142 countries/regions from 2008 to 2019.

## Empirical results

### Baseline regression

In this paper, the TERGM model is estimated and fitted for the international trade relationship network from 2008 to 2019. Since the R-squared statistic is difficult to assess the true effects of the TERGM model regression, the comparison of the Area Under the Curve of the Receiver Operating Characteristic (AUC-ROC) and the Area Under the Curve of the Precision-Recall (AUC-PR) for observed networks and random networks is used as a reference for the model's fitting performance, as shown in Table 2. Column (1) represents the baseline model controlling only node attributes, column (2) adds exogenous mechanism variables to the baseline, and column (3) is the comprehensive test model with the inclusion of endogenous mechanism variables.

Regarding the explanatory variables, as indicated by models (1) to (3), the coefficients of institutional distance are consistently negative and statistically significant. This suggests that a reduction in institutional distance increases the probability of forming intellectual property trade relationships, aligning with the theoretical analysis and validating H1.

In the examination of the exogenous mechanisms, adjacency network, language network, and colonial network all exhibit significant positive correlations with the formation of intellectual property trade networks. This implies that economies that share geographical borders, use a common language, or have a colonial history are more likely to establish intellectual property trade relationships. Regarding the endogenous mechanism variables, we primarily measure mutual, triple, transitivity, and cyclicality, as well as outdegree and indegree. In binary relations, mutual exhibits a positive and significant correlation, suggesting that one-way trade relationships between economies tend to develop into two-way trade relationships. In the context of triadic relations, both transitivity and cyclicality are significantly correlated, with the former having a positive coefficient and the latter a negative coefficient. This outcome indicates a hierarchical clustering pattern in the formation of intellectual property trade networks. Furthermore, in terms of outdegree and indegree, as demonstrated by model (3), outdegree has a positive and significant coefficient, indicating a substantial expansion in the intellectual property trade of the exporting countries, while indegree has a negative and significant coefficient, signifying a divergent trend in the importing nodes of the exporting countries.

### Goodness of fit tests

In assessing the Goodness of fit (GOF) for the TERGM model, we primarily employ the GOF process. This involves simulating networks based on the model parameters and comparing these simulated networks with the real network to analyze the model's fitting performance. This paper conducts 100 simulations using the parameters of the comprehensive test model, which includes both endogenous and exogenous relationships, as represented by TERGM model (3). The data is subjected to logarithmic transformation for visualization.

**Table 2. Baseline regression.**

| Intellectual Property Trade Network | | (1) | (2) | (3) |
|---|---|---|---|---|
| Explanatory Variables | Institutional distance | -0.0038*** (0.0006) | -0.0034*** (0.0006) | -0.0006*** (0.0002) |
| Node Attribute Variables | E-Exports of Services | -0.0636*** (0.0132) | -0.0617*** (0.0155) | 0.0041 (0.0109) |
| | I- Exports of Services | 0.6336*** (0.0728) | 0.6486*** (0.0772) | -0.1083*** (0.009) |
| | E-Exports of Goods | 0.0017 (0.0093) | 0.0059 (0.0115) | -0.0041 (0.0159) |
| | I- Exports of Goods | -0.262 (0.1661) | -0.2653 (0.1732) | 0.0072 (0.0128) |
| | E-GDP | -0.0083 (0.0098) | -0.0198* (0.0127) | -0.0182 (0.0183) |
| | I-GDP | 0.6927** (0.2185) | 0.6921** (0.2268) | 0.0131 (0.0122) |
| | E-FDI | 0.0034** (0.0012) | 0.0025 (0.0012) | -0.0001 (0.0019) |
| | I-FDI | 0.0123 (0.0085) | 0.0111 (0.0085) | 0.0089*** (0.0016) |
| Exogenous Variables | Border network | | 1.5718*** (0.0854) | 2.0579*** (0.0766) |
| | Language network | | 0.3434*** (0.0572) | 0.6233*** (0.0349) |
| | Colonial network | | 0.6041*** (0.0329) | 0.6022*** (0.0428) |
| Endogenous Variables | Mutual | | | 1.0738*** (0.0416) |
| | Ttriple | | | 0.0047*** (0.0003) |
| | Transitiveties | | | 0.2485* (0.108) |
| | Cyclicalties | | | -0.1063*** (0.0081) |
| | Out-2-stars | | | 0.1355*** (0.0012) |
| | In-2-stars | | | -0.0311*** (0.0024) |
| | Outdegree | | | 1.7472*** (0.0347) |
| | Indegree | | | -0.0972*** (0.0063) |
| Edges | | -31.2492*** (0.6113) | -31.5105*** (0.7402) | -13.9008*** (0.4477) |
| ROC/Random | | 0.9478/0.5070 | 0.9491/0.5117 | 0.9680/0.5109 |
| PR/Random | | 0.8707/0.2173 | 0.8730/0.2201 | 0.9217/0.2204 |
| Observations | | 143×142×12 | | |

*Notes*: The dependent variables are all related to the intellectual property export network. Following established research practices, this study further computes more precise p-values. Symbols

\*, \*\*, and \*\*\* denote significance at the 5%, 1%, and 0.1% levels, respectively. Standard deviations are provided in parentheses; the format is consistent in the following tables.

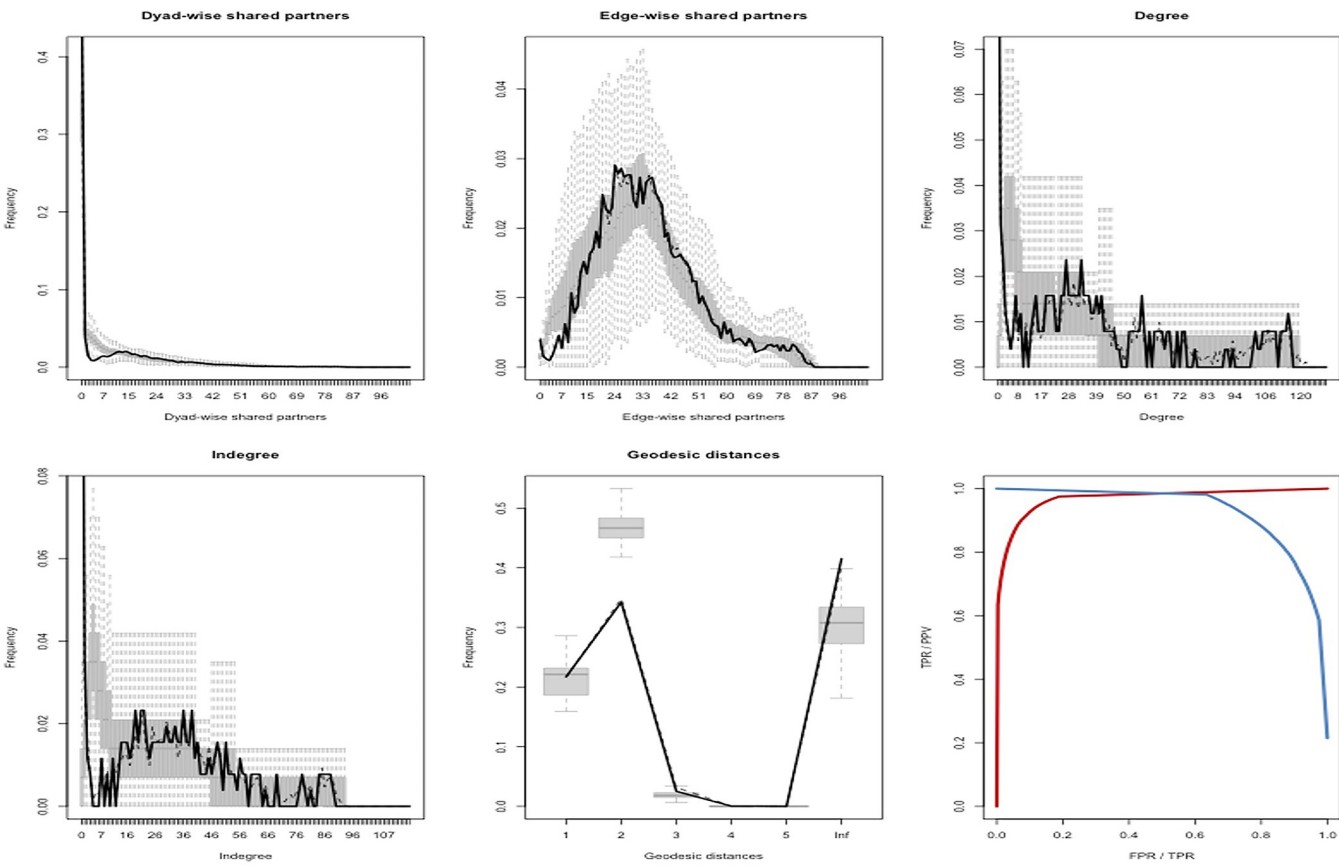

**Fig 1. Goodness of fit results.**

In the Fig 1, The first five subplots depict a comparison of various network feature indicators. These include dyad-wise shared partners, edge-wise shared partners, geodesic distances, degree centrality, and indegree centrality. The black solid lines represent the results from the observed international trade relations network, while the gray area represents the measurements of the simulated network within a 95% confidence interval, following the approach by Harris [35]. By examining the graphs, it is evident that the simulated network effectively represents the structural characteristics of the observed intellectual property trade relations network. In the last subplot, the red ROC curve represents the proportion of international trade relations that exist simultaneously in both the simulated and observed networks. The results show that as the number of simulations increases, this proportion gradually approaches 1, indicating a strong model fitting performance.

## Robustness tests

### Year cross-section analysis

To further confirm the inhibitory effect of institutional distance on the intellectual property trade network, this study conducted yearly regressions on 30 countries with relatively high levels of institutional quality. This approach aimed to observe how the impact of institutional distance varies in the global intellectual property network. Additionally, network density's influence was considered. The results of the yearly regression on institutional distance and intellectual property trade volume are presented in Table 3. Firstly, the coefficients for

**Table 3. Subsample yearly regression results.**

| Year | 2008 | 2009 | 2010 | 2011 | 2012 | 2013 |
|---|---|---|---|---|---|---|
| Institutional distance | -0.0315*** (0.0073) | -0.0348*** (0.0072) | -0.0383*** (0.008) | -0.0528*** (0.0103) | -0.0496*** (0.0108) | -0.0527*** (0.0122) |
| Control Variables | Yes | Yes | Yes | Yes | Yes | Yes |
| Observations | 143×142 | 143×142 | 143×142 | 143×142 | 143×142 | 143×142 |
| Year | 2014 | 2015 | 2016 | 2017 | 2018 | 2019 |
| Institutional distance | -0.0630*** (0.0122) | -0.0562*** (0.0126) | -0.0589*** (0.0131) | -0.0754*** (0.0147) | -0.0614*** (0.0156) | -0.0737*** (0.0154) |
| Control Variables | Yes | Yes | Yes | Yes | Yes | Yes |
| Observations | 143×142 | 143×142 | 143×142 | 143×142 | 143×142 | 143×142 |

institutional distance remain consistently negative and significant, once again confirming the conclusion of Hypothesis 1. Secondly, it can be observed that the coefficients for institutional distance increase annually, expanding from -0.0315 in 2008 to -0.0737 in 2019, a 57.26% increase. The average annual expansion rate is 9.11%. This indicates that the influence of institutional distance on intellectual property trade is gradually strengthening. As economic development and global dynamics evolve, countries are becoming more sensitive to institutional distance, and the requirements for institutional alignment in intellectual property trade are increasing. This underscores the importance for nations and international organizations to recognize the growing impact of institutional distance on intellectual property trade and to strengthen international cooperation for the promotion of intellectual property protection and institutional coordination.

## Sub-indicator regression

The paper begins by regressing the six sub-indices that constitute institutional distance to provide a more detailed exploration of each sub-index's impact on intellectual property trade. The regression results are presented in Table 4.

The table shows that the coefficient for "Voice and Accountability" (VA) is 0.001 and significant, indicating that differences among countries in terms of voice and accountability have a positive impact on intellectual property trade relationships. This suggests that a higher degree of dialogue and cooperation among countries in the context of intellectual property trade can lead to more easily established intellectual property exchanges.

On the other hand, the coefficients for "Rule of Law" (RL) and "Control of Corruption" (CC) are -0.0006 and -0.0003, respectively, although not statistically significant. This may

**Table 4. Results of sub-index regressions.**

| Dependent Variable | Intellectual Property Trade Network | | | | | |
|---|---|---|---|---|---|---|
| Sub-Index | VA | PSNV | GE | RQ | RL | CC |
| | 0.0010*** (0.0002) | -0.0019*** (0.0005) | -0.0012** (0.0005) | -0.0039*** (0.0002) | -0.0006 (0.0004) | -0.0003 (0.0004) |
| ROC Random | 0.9680 0.5077 | 0.9677 0.5103 | 0.9667 0.5112 | 0.9686 0.5092 | 0.9673 0.5077 | 0.9686 0.5067 |
| PR Random | 0.9210 0.2186 | 0.9215 0.2202 | 0.9192 0.2201 | 0.9226 0.2187 | 0.9211 0.2184 | 0.9227 0.2178 |
| Control Variables | Yes | Yes | Yes | Yes | Yes | Yes |
| Observations | 143×142×12 | | | | | |

indicate that the institutional distance in these two aspects is not prominent enough, or other factors may be interfering with their impact. In a study by Chen [12], she analyzed the influence of sub-indices of institutional distance on the parent network of intellectual property trade, which is the service trade network. The results showed that the distance related to control of corruption was negative but not significant, consistent with the results of this paper. The distance related to the rule of law was negative and significant, suggesting that the control of corruption distance might not significantly affect the intellectual property trade network, while the rule of law, as a crucial factor in intellectual property trade, involving dispute arbitration and intellectual property protection, cannot be adequately measured by distance. More often, it's the relative level of legal systems that influences the formation of the intellectual property trade network.

The coefficients for other sub-indices are all negative and significant, indicating that differences in institutional distance in these aspects have a negative impact on intellectual property trade, further confirming H1. These results are consistent with the conclusions of the baseline model, emphasizing the comprehensive impact of institutional distance on intellectual property trade.

## Multidimensional fixed effects

The paper further validates its findings by using a multidimensional fixed-effects model to examine the fixed effects across three dimensions: exporting and importing countries and years. This analysis controls for the export and import control variables. The results, as presented in Table 5, demonstrate that countries with smaller institutional distances are more likely to establish intellectual property trade relationships, consistent with the conclusions of the baseline model. The application of a multidimensional fixed-effects model contributes to a more comprehensive understanding of the impact of institutional distance on intellectual property trade. By controlling for fixed effects across multiple dimensions, we can more accurately capture differences among countries and years, leading to more reliable conclusions. The results indicate that even when considering other factors, countries with smaller institutional distances are still more likely to establish intellectual property trade relationships, further confirming the conclusions of H1.

## Mechanism analysis

This paper posits that institutional distance affects the formation of intellectual property trade networks through its impact on the signing of trade agreements. To validate this mediating

**Table 5. Results of multidimensional fixed-effects regression.**

| Intellectual Property Trade Network | (1) | (2) | (3) |
|---|---|---|---|
| Institutional distance | -0.000480*** (0.000) | -0.000479*** (0.000) | -0.000482*** (0.000) |
| E- Control Variables | | Yes | Yes |
| I- Control Variables | | | Yes |
| Year Fixed effect | Yes | Yes | Yes |
| E-countries Fixed effect | Yes | Yes | Yes |
| I- countries Fixed effect | Yes | Yes | Yes |
| Observations | 143×142×12 | 143×142×12 | 143×142×12 |
| $R^2$ | 0.5441 | 0.5442 | 0.5442 |
| Adjusted-$R^2$ | 0.5435 | 0.5436 | 0.5436 |

**Table 6. Analysis of the mediating effect of trade agreements.**

| Variables | Trade agreement | Intellectual Property Trade Network | Intellectual Property Trade Network |
|---|---|---|---|
| Institutional distance | -0.0022*** (0.0004) | | -0.0019*** (0.0005) |
| Trade agreement | | 0.578*** (0.1096) | 0.5668*** (0.1073) |
| Control Variables | Yes | Yes | Yes |
| ROC/Random | 0.6112/0.5256 | 0.9033/0.5078 | 0.9031/0.5145 |
| PR/Random | 0.2439/0.1822 | 0.7807/0.2181 | 0.7820/0.2222 |
| Observations | 143×142×12 | 143×142×12 | 143×142×12 |

effect, an intermediate effect model is constructed based on the baseline model (1). In this model, once a trade agreement is formed in year "t," it remains in the "formed" state in each subsequent year. However, this post-formation state in subsequent years is the result of the formation of a trade agreement in a previous year, rather than the contracting parties reselecting the agreement each year. This poses a challenge related to the issue of serial correlation in the formation of trade agreements. The model in this paper, based on Hanneke [39], employs a discrete-time social network model, dividing time into discrete time steps and considering the probability of node changes using Markov simulation. This overcomes the issue of serial correlation. After controlling for the node attributes of exporting and importing countries, the results are shown in Table 6.

The results reveal that the coefficient of institutional distance on trade agreements is -0.0022 and statistically significant, indicating that smaller institutional distance promotes the signing of trade agreements. The coefficient of trade agreements on intellectual property trade networks is 0.578 and statistically significant, suggesting that trade agreements facilitate the formation of intellectual property trade networks. After incorporating the trade agreement variables, the impact coefficient of institutional distance on intellectual property trade networks decreases from -0.0038 in the baseline model (1) to -0.0019 in Column (3). This implies that the negative influence of institutional distance on two countries that have signed a trade agreement becomes smaller. Further calculations reveal that the mediating effect of trade agreements on institutional distance and intellectual property networks is 33.46%. Thus, H2 is confirmed. Chen [12], introduced an interaction term between institutional distance and trade agreements to study the moderating effect of trade agreements on institutional distance and service trade. The results showed that the coefficients of the sub-indicators were either positive or negative and not statistically significant. This further emphasizes that trade agreements do not directly promote the impact of institutional distance on intellectual property trade networks but rather achieve it through their mediating mechanism. The signing of trade agreements changes the level of knowledge protection, regulatory systems, tariffs, and other factors between countries [1], and establishes more comprehensive dispute resolution mechanisms, thus promoting the formation of intellectual property trade networks.

## Conclusion and policy recommendations

Based on data from the intellectual property trade network of 143 countries from 2008 to 2019, this paper employs a time-series Exponential Random Graph Model (TERGM) to examine the relationship between institutional distance and the intellectual property trade network. The research results indicate that a smaller institutional distance can reduce transaction costs and contract risks, making it easier for the two countries to engage in intellectual property

trade. After controlling for factors such as geographical proximity, shared language, colonial relations, and self-organizing characteristics within the intellectual property trade network, the conclusion that reducing institutional distance promotes intellectual property trade still holds. The results of annual regression tests indicate that the impact of institutional distance on the intellectual property trade network fluctuates and increases over time. Research on the sub indicators of distance has found that the smaller the institutional distance in terms of political stability, government efficiency, and regulatory quality, the more likely it is to establish intellectual property trade relationships. Mechanism analysis shows that the institutional distance between the two countries will affect the establishment of intellectual property trade networks by influencing the signing of trade agreements between the two countries.

Based on the above conclusions, the recommendations are as follows: Firstly, actively reducing institutional distance with high-level institutional countries is conducive to developing intellectual property trade relations, according to the conclusions drawn in this paper. Therefore, under the premise of actively improving and developing the domestic institutional system, countries should align with the global development pattern, actively integrate into international economic and trade rules, continuously improve their intellectual property protection levels, promote domestic economic development and innovation, and gain long-term advantages in international resource allocation and participation in global division of labor. Secondly, attaching importance to intellectual property trade with countries at similar institutional levels within the region. Based on the research findings of this paper, the impact of institutional distance is expanding year by year. Therefore, countries should pay more attention to domestic institutional reforms. Combining the self-organizing characteristics of intellectual property trade networks, they should further focus on regional economic trade, promote the process of regional economic integration, and strive to expand the marginal of intellectual property trade networks. Efficiently utilize the international market space and trade cooperation relations with neighboring countries to reduce trade costs and improve market efficiency. Thirdly, actively promoting the signing and implementation of free trade agreements. According to the research findings of this paper, countries with smaller institutional distances are more likely to sign trade agreements, thereby promoting intellectual property trade between the two countries. It is conducive to the targeted promotion of the development of domestic institutional construction, facilitating the attraction of trade agreement relations with more countries, and maximizing the trade promotion effects brought about by such agreements. Fourthly, there is a possibility for countries to increase institutional distance in intellectual property trade by signing trade agreements. The conclusion of this article suggests that institutional distance may promote intellectual property trade through trade agreements. In the context of hindered globalization, bilateral or multilateral trade agreements can promote intellectual property trade between countries.

This paper makes significant theoretical contributions to shifting the traditional perspectives on intellectual property trade research. Firstly, by focusing on the formation and evolution of dyads and triads within trade networks, this study innovatively examines how institutional distances influence the embeddedness patterns in intellectual property trade networks, offering new mechanistic explanations for understanding the structural characteristics of trade relationships. Secondly, by adopting a global perspective on intellectual property trade networks and conducting in-depth analyses of the potential influences of importing and exporting countries, external scenarios, and internal dependency relationships, it achieves more comprehensive and robust research conclusions. Furthermore, the application of the Exponential Random Graph Model provides a powerful tool for analyzing and simulating the structural adjustments and improvements of intellectual property trade networks, capable of integrating node attributes with various internal and external mechanisms and influences.

Ultimately, the research framework, by integrating the dynamics of power relations, geopolitical strategies, and economic interests, delves into the complex interactions between institutional distances, intellectual property trade networks, and trade agreements, offering profound theoretical insights from a socio-political dimension. This series of theoretical contributions not only broadens the boundaries of intellectual property trade research but also provides a theoretical basis and practical guidance for related policy formulation. Due to the intangible and difficult-to-measure nature of intellectual property trade, obtaining bilateral trade volume data poses challenges. Therefore, this study has limitations due to incomplete data and a relatively short time span in the empirical process. Future research could consider refining the impact mechanisms of formal and informal institutional distance on both narrow and broad measures of intellectual property trade volume to gain a more comprehensive understanding of the complex relationship between intellectual property protection and international trade. Such research would help delve deeper into the underlying mechanisms of intellectual property trade under different institutional environments, providing policymakers with more targeted policy recommendations to promote win-win development in intellectual property protection and international trade.

## Supporting information

**S1 Data. Data obtained through public resource has been uploaded as a supporting information file, DOI: 10.57760/sciencedb.09553.**
(XLSX)

**S2 Data.**
(XLSX)

**S3 Data.**
(XLSX)

**S4 Data.**
(XLSX)

## Acknowledgments

We thank those anonymous reviewers and the editor whose comments/suggestions helped improve and clarify this manuscript.

## Author Contributions

**Conceptualization:** Jiangjiao Wang.

**Data curation:** Yida Wang.

**Funding acquisition:** Jiangjiao Wang.

**Methodology:** Jiangjiao Wang.

**Software:** Yida Wang.

**Supervision:** Jiangjiao Wang.

**Writing – original draft:** Yida Wang.

**Writing – review & editing:** Jiangjiao Wang.

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
