## [Decision Letter · Decision Letter 0]

22 Apr 2024

PONE-D-23-37149Institutional Distance, Trade Agreements, and Intellectual Property Trade Networks: Evidence from Cross-Border DataPLOS ONE

Dear Dr. Li,

Thank you for submitting your manuscript to PLOS ONE. After careful consideration, we feel that it has merit but does not fully meet PLOS ONE’s publication criteria as it currently stands. Therefore, we invite you to submit a revised version of the manuscript that addresses the points raised during the review process.

We look forward to receiving your revised manuscript.

Kind regards,

Zahra Masood Bhutta

Academic Editor

PLOS ONE

Journal Requirements:

4. Please ensure that you refer to Figure 1 in your text as, if accepted, production will need this reference to link the reader to the figure.

Reviewers' comments:

Reviewer's Responses to Questions

**Comments to the Author**

1. Is the manuscript technically sound, and do the data support the conclusions?

Reviewer #1: Yes

Reviewer #2: Partly

2. Has the statistical analysis been performed appropriately and rigorously? 

Reviewer #1: Yes

Reviewer #2: I Don't Know

3. Have the authors made all data underlying the findings in their manuscript fully available?

Reviewer #1: Yes

Reviewer #2: Yes

4. Is the manuscript presented in an intelligible fashion and written in standard English?

Reviewer #1: Yes

Reviewer #2: Yes

5. Review Comments to the Author

Reviewer #1: This is good written paper that deserves publication. The dataset and the aim of the paper should be clarified in detail. On the other hand, why those years are used for data collection? The literature review can be enriched by seeking most recent papers.

Reviewer #2: 1. The concept is not clearly defined in the manuscript. What are the definitions of institutional distance, intellectual property right trade network, and trade agreements?

2. The manuscript provides an adequate description of the relevant literature, but the discourse of existing research gaps is simplistic and not clear and specific. What is the theoretical significance of the study compared with previous studies?

3. The practical enlightenment is too general and doesn’t have pertinence.

4. What are the limitations and future research directions?

5. The manuscript also has some errors in the grammar and details. For example, the number and order of the Tables are confused.

6. Is it “intellectual property trade network” or “intellectual property right trade network”?

7. The manuscript doesn’t pay enough attention to the latest research, so it is recommended to add citations of literature within 5 years.

---

## [Author Response · Author response to Decision Letter 0]

8 May 2024

Response to Reviewers

First of all, we would like to thank reviewers for your insightful, constructive, and helpful comments on our manuscript entitled “Institutional Distance, Trade Agreements, and Intellectual Property Trade Networks: Evidence from Cross-Border Data” We have carefully considered and addressed all the comments and made necessary revisions in the revised manuscript. We provide a point-by-point response to the reviewers’ comments below.

The points raised by the reviewers are written in bold font, whereas our responses are shown in normal font, and the quotation of the revised manuscript is shown in red font. 

Reviewer #1: 

1. Reviewer #1: This is good written paper that deserves publication. The dataset and the aim of the paper should be clarified in detail. On the other hand, why those years are used for data collection? The literature review can be enriched by seeking most recent papers. 

Response:

Thank you for your recognition of our paper. Of course, the issue you mentioned is also a problem that exists in this article. We accept unconditionally. In revising the manuscript, we supplemented the data description and reorganized the literature section, adding the latest literature. The revised parts are as follows:

…

Institutional distance primarily refers to the differences in formal institutions among countries, including legal systems, political stability, government efficiency, regulatory quality, and other aspects. Good institutions can effectively allocate resources and reduce transaction costs, while poorer institutional environments lack protection for transactional rationality, thus dampening trading enthusiasm. The intellectual property trade network is a multilateral trade network centered around intellectual property rights, primarily encompassing activities such as licensing and transfer of intellectual property. These transactions are characterized by heterogeneity, intangibility, and complexity, making the influence of institutional environments more direct and crucial. Countries with high-level institutional environments generally exhibit higher intellectual property output. However, when engaging in trade with countries with low intellectual property output but also low-level institutional environments, the transactional risks are heightened. The relationship between the two has sparked controversy. Thus, does institutional distance influence the network of intellectual property trade? If there is an influence, what is the mechanism of this impact? Studying the relationship between institutional distance and the network of intellectual property trade, as outlined in the previous question, can provide insights and policy implications for reconciling global intellectual property conflicts, advancing institutional reforms in various countries, and reducing risks in international intellectual property trade.

At present, there is limited academic research on the impact of institutional distance on the intellectual property trade network, but there is a more extensive discussion on both separately. In the terms of intellectual property trade, with the development of the intellectual economy, the intellectual property market has become a crucial foundation for a country to promote innovation and enhance its national trade competitiveness [1]. The maturity of a country's market institutions significantly affects the willingness of host countries to import their intellectual property [2]. Due to the contractual nature of intellectual property trade, institutional elements inevitably become the focus of research, among which the level of intellectual property protection is widely scrutinized by domestic and international scholars. Enhancing the level of intellectual property protection can effectively raise a country's innovation level and improve the structure of intellectual property trade [3]. In terms of institutional distance, since [4] first introduced the concept of institutional distance, numerous scholars have engaged in discussions on the relationship between institutional distance and international trade: Based on the New-New Trade Theory and New Institutional Economics, overall institutional quality can positively impact international trade through means such as transaction costs and incomplete contracts [5]. From an economic institutional perspective, the average institutional distance constructed can affect foreign trade through the suppression of transaction costs [6]. From the viewpoint of New Institutional Economics, formal and informal institutional distances have different effects on service trade [7]; From the perspective of multinational enterprises, institutional distance significantly affects the performance of overseas subsidiaries [8]. In the context of bidirectional talent flow in technology between countries, institutional distance plays a significant role [9]. Different levels of institutional distance have varied impacts on economic activities; countries' outward investments exhibit characteristics of "political institutional proximity" and "economic institutional escape" [10]. Another portion of the literature provides evidence of the role of trade agreements in promoting intellectual property trade. Trade agreements that include levels of intellectual property protection can facilitate inter-country intellectual property trade and promote the upgrading of their industrial chains [11]; Regional trade agreements can enhance the facilitating role of institutional quality in intellectual property trade [12]; Regional trade agreements can help bridge institutional gaps and mitigate the impact of bilateral institutional distance [13]. For developing countries, the intellectual property clauses in trade agreements may lead to short-term losses due to high patent fees and lower returns [14]. There is also an analysis based on mega trade agreements such as RCEP, which offers recommendations for countries participating in the formulation of rules for intellectual property trade [15]. Regional trade agreements can enhance the facilitating role of institutional quality in intellectual property trade [12]. In terms of research methods, most scholars examined the probability of institutional distance and trade relationships based on traditional trade gravity models and spatial models, including discussions from various perspectives such as adjacent effects and national heterogeneity, the impact intensity of various institutions [16], controlling for culture and geographic distance [17]. However, as the intellectual property rights trade network is a highly important part of international trade networks, with the development of technology and information technology, traditional spatial and gravity models may find it difficult to detect the internal interdependencies and embedded external relationships. Overall, existing literature on the effects of institutional distance on various types of international economic activities is abundant, demonstrating the significant impact of institutional distance in international trade research. However, it still fails to validate the specific effects of institutional distance on intellectual property trade. Moreover, research on intellectual property trade networks is limited, and most existing literature relies on traditional trade gravity models, making it difficult to explore the network characteristics of intellectual property trade. Consequently, it is challenging to explain the forms of intellectual property trade between countries with different institutional distances in reality. Furthermore, in the international intellectual property trade landscape reshaped by multilateral trade agreements, the aforementioned studies still offer limited insights into understanding institutional distance, trade agreements, and intellectual property trade networks.

···

Data Sources

The data for this study primarily come from the following sources: First, import and export data are sourced from the World Trade Organization-Organization for Economic Co-operation and Development Services Trade Balance Dataset (BaTIS). After threshold conversion, a binary matrix of intellectual property trade for 143×142 countries/regions is obtained. Second, institutional distance is derived from the World Bank's Global Governance Indicators, calculated based on six sub-indicators to generate a 143×142 matrix of institutional distance. Third, exogenous relationship networks, including adjacency networks, language networks, and colonial networks, are compiled from the CEPII database. Fourth, the bilateral trade agreement network comes from the Design of Trade Agreements (DESTA) database at the University of Bern. Fifth, other control variables are sourced from publicly available World Bank data. To mitigate the influence of the 2008 financial crisis and the 2019 global public health emergency on this study, the final dataset comprises intellectual property trade network data for 143×142 countries/regions from 2008 to 2019.

···

Reviewer #2: 

1. The concept is not clearly defined in the manuscript. What are the definitions of institutional distance, intellectual property right trade network, and trade agreements?

Response:

Thank you very much for your valuable feedback. The original manuscript lacked clear definitions of concepts related to institutional distance, intellectual property trade networks, and trade agreements, leading to ambiguity in the direction of the study. We have supplemented the introduction with explanations of these concepts and their relationships. The revised parts are as follows:

… 

Institutional distance primarily refers to the differences in formal institutions among countries, including legal systems, political stability, government efficiency, regulatory quality, and other aspects. Good institutions can effectively allocate resources and reduce transaction costs, while poorer institutional environments lack protection for transactional rationality, thus dampening trading enthusiasm. The intellectual property trade network is a multilateral trade network centered around intellectual property rights, primarily encompassing activities such as licensing and transfer of intellectual property. These transactions are characterized by heterogeneity, intangibility, and complexity, making the influence of institutional environments more direct and crucial. Countries with high-level institutional environments generally exhibit higher intellectual property output. However, when engaging in trade with countries with low intellectual property output but also low-level institutional environments, the transactional risks are heightened. The relationship between the two has sparked controversy. Thus, does institutional distance influence the network of intellectual property trade? If there is an influence, what is the mechanism of this impact? Studying the relationship between institutional distance and the network of intellectual property trade, as outlined in the previous question, can provide insights and policy implications for reconciling global intellectual property conflicts, advancing institutional reforms in various countries, and reducing risks in international intellectual property trade.

…

2. The manuscript provides an adequate description of the relevant literature, but the discourse of existing research gaps is simplistic and not clear and specific. What is the theoretical significance of the study compared with previous studies?

Response:

Thank you very much for your valuable feedback. We have reorganized past relevant literature, identified some crucial theoretical gaps, and elaborated on the significance of our study compared to previous studies. The revised parts are as follows:

…

At present, there is limited academic research on the impact of institutional distance on the intellectual property trade network, but there is a more extensive discussion on both separately. In the terms of intellectual property trade, with the development of the intellectual economy, the intellectual property market has become a crucial foundation for a country to promote innovation and enhance its national trade competitiveness [1]. The maturity of a country's market institutions significantly affects the willingness of host countries to import their intellectual property [2]. Due to the contractual nature of intellectual property trade, institutional elements inevitably become the focus of research, among which the level of intellectual property protection is widely scrutinized by domestic and international scholars. Enhancing the level of intellectual property protection can effectively raise a country's innovation level and improve the structure of intellectual property trade [3]. In terms of institutional distance, since first introduced the concept of institutional distance, numerous scholars have engaged in discussions on the relationship between institutional distance and international trade [4]; Based on the New-New Trade Theory and New Institutional Economics, overall institutional quality can positively impact international trade through means such as transaction costs and incomplete contracts [5]. From an economic institutional perspective, the average institutional distance constructed can affect foreign trade through the suppression of transaction costs [6]. From the viewpoint of New Institutional Economics, formal and informal institutional distances have different effects on service trade [7]; From the perspective of multinational enterprises, institutional distance significantly affects the performance of overseas subsidiaries [8]. In the context of bidirectional talent flow in technology between countries, institutional distance plays a significant role [9]. Different levels of institutional distance have varied impacts on economic activities; countries' outward investments exhibit characteristics of "political institutional proximity" and "economic institutional escape" [10]. Another portion of the literature provides evidence of the role of trade agreements in promoting intellectual property trade. Trade agreements that include levels of intellectual property protection can facilitate inter-country intellectual property trade and promote the upgrading of their industrial chains [11]; Regional trade agreements can enhance the facilitating role of institutional quality in intellectual property trade [12]; Regional trade agreements can help bridge institutional gaps and mitigate the impact of bilateral institutional distance [13]. For developing countries, the intellectual property clauses in trade agreements may lead to short-term losses due to high patent fees and lower returns [14]. There is also an analysis based on mega trade agreements such as RCEP, which offers recommendations for countries participating in the formulation of rules for intellectual property trade [15]. Regional trade agreements can enhance the facilitating role of institutional quality in intellectual property trade [12]. In terms of research methods, most scholars examined the probability of institutional distance and trade relationships based on traditional trade gravity models and spatial models, including discussions from various perspectives such as adjacent effects and national heterogeneity, the impact intensity of various institutions [16], controlling for culture and geographic distance [17]. However, as the intellectual property rights trade network is a highly important part of international trade networks, with the development of technology and information technology, traditional spatial and gravity models may find it difficult to detect the internal interdependencies and embedded external relationships. Overall, existing literature on the effects of institutional distance on various types of international economic activities is abundant, demonstrating the significant impact of institutional distance in international trade research. However, it still fails to validate the specific effects of institutional distance on intellectual property trade. Moreover, research on intellectual property trade networks is limited, and most existing literature relies on traditional trade gravity models, making it difficult to explore the network characteristics of intellectual property trade. Consequ

---

## [Decision Letter · Decision Letter 1]

21 Jun 2024

PONE-D-23-37149R1Institutional Distance, Trade Agreements, and Intellectual Property Trade Networks: Evidence from Cross-Border DataPLOS ONE

Dear Dr. Li,

Thank you for submitting your manuscript to PLOS ONE. After careful consideration, we feel that it has merit but does not fully meet PLOS ONE’s publication criteria as it currently stands. Therefore, we invite you to submit a revised version of the manuscript that addresses the points raised during the review process.

We look forward to receiving your revised manuscript.

Kind regards,

Zahra Masood Bhutta

Academic Editor

PLOS ONE

Journal Requirements:

Reviewers' comments:

Reviewer's Responses to Questions

**Comments to the Author**

1. If the authors have adequately addressed your comments raised in a previous round of review and you feel that this manuscript is now acceptable for publication, you may indicate that here to bypass the “Comments to the Author” section, enter your conflict of interest statement in the “Confidential to Editor” section, and submit your "Accept" recommendation.

Reviewer #1: All comments have been addressed

Reviewer #2: (No Response)

2. Is the manuscript technically sound, and do the data support the conclusions?

Reviewer #1: Yes

Reviewer #2: Yes

3. Has the statistical analysis been performed appropriately and rigorously? 

Reviewer #1: Yes

Reviewer #2: Yes

4. Have the authors made all data underlying the findings in their manuscript fully available?

Reviewer #1: Yes

Reviewer #2: Yes

5. Is the manuscript presented in an intelligible fashion and written in standard English?

Reviewer #1: Yes

Reviewer #2: Yes

6. Review Comments to the Author

Reviewer #1: (No Response)

Reviewer #2: The author should explain the theoretical contribution in the conclusion.

The article still needs attention to the details.

7. PLOS authors have the option to publish the peer review history of their article (what does this mean?). If published, this will include your full peer review and any attached files.

Reviewer #1: No

Reviewer #2: No

---

## [Author Response · Author response to Decision Letter 1]

22 Jun 2024

Response to Reviewers

First of all, we would like to thank reviewers for your insightful, constructive, and helpful comments on our manuscript entitled “Institutional Distance, Trade Agreements, and Intellectual Property Trade Networks: Evidence from Cross-Border Data” We have carefully considered and addressed all the comments and made necessary revisions in the revised manuscript. We provide a point-by-point response to the reviewers’ comments below.

The points raised by the reviewers are written in bold font, whereas our responses are shown in normal font, and the quotation of the revised manuscript is shown in red font. 

Reviewer #2: 

1. The author should explain the theoretical contribution in the conclusion.The article still needs attention to the details.

Response:

Thank you very much for your valuable feedback.In the revised version, we have included the content of the modified theoretical contributions in the conclusion section and made refinements to some details throughout the document.

Introduction

With the advent of the knowledge economy era, intellectual property trade based on cross-border transfer or licensing of intellectual property rights gradually became one of the three major trade in the world. As of 2017, intangible capital such as intellectual property created 30.4% of the value of global manufactured goods trade, determining the success rate of products in the market and serving as a strategic resource for the country to improve its core competitiveness. The creation of intellectual property has evolved from being monopolized by highly developed countries in the past to moving towards a global value chain division of labor. In recent years, the global economy is in a prolonged state of stagnation, leading to escalating political and economic tensions among nations, resulting in a growing frequency of trade frictions related to intellectual property. 

TRIPs and the WTO have had a huge impact as multilateral trade dispute resolution mechanisms, but in the new situation, the traditional TRIPs and WTO-centered multilateral negotiations gradually lose their impact, while the various institutional requirements for the development of intellectual property trade become increasingly stringent and complex. This trend pushes the evolution of the intellectual property network toward two extremes: on one hand, developed countries seek to maintain their dominance in the global trade value chain by continuously raising the standards of intellectual property protection, which actively advocate for the establishment of multilateral, bilateral, and regional international agreements, aiming to elevate their national policies to global standards, thereby expanding their intellectual property policies to maximize their national interests on a global scale. On the other hand, developing countries and certain civil society forces are committed to building a more equitable framework for intellectual property protection. They aim to break away from past global trade agreements with high levels of intellectual property protection and, instead, seek bilateral or multilateral trade agreements while formulating policies that are more inclusive and sustainable, all with the objective of ensuring the stability and fairness of intellectual property trade. Among the many factors that affect the construction of bilateral intellectual property trade networks, institutional differences are receiving increasing attention. Institutional distance may result in multinational enterprises facing varying intellectual property environments in different countries, increasing compliance costs and legal risks. This not only significantly impacts a firm's global strategy and competitiveness but also underscores the international community's growing emphasis on global coordination and consistency in intellectual property rights. 

Institutional distance primarily refers to the differences in formal institutions among countries, including legal systems, political stability, government efficiency, regulatory quality, and other aspects. Good institutions can effectively allocate resources and reduce transaction costs, while poorer institutional environments lack protection for transactional rationality, thus dampening trading enthusiasm. The intellectual property trade network is a multilateral trade network centered around intellectual property rights, primarily encompassing activities such as licensing and transfer of intellectual property. These transactions are characterized by heterogeneity, intangibility, and complexity, making the influence of institutional environments more direct and crucial. Countries with high-level institutional environments generally exhibit higher intellectual property output. However, when engaging in trade with countries with low intellectual property output but also low-level institutional environments, the transactional risks are heightened. The relationship between the two has sparked controversy. Thus, does institutional distance influence the network of intellectual property trade? If there is an influence, what is the mechanism of this impact? Studying the relationship between institutional distance and the network of intellectual property trade, as outlined in the previous question, can provide insights and policy implications for reconciling global intellectual property conflicts, advancing institutional reforms in various countries, and reducing risks in international intellectual property trade.

At present, there is limited academic research on the impact of institutional distance on the intellectual property trade network, but there is a more extensive discussion on both separately. In the terms of intellectual property trade, with the development of the intellectual economy, the intellectual property market has become a crucial foundation for a country to promote innovation and enhance its national trade competitiveness [1]. The maturity of a country's market institutions significantly affects the willingness of host countries to import their intellectual property [2]. Due to the contractual nature of intellectual property trade, institutional elements inevitably become the focus of research, among which the level of intellectual property protection is widely scrutinized by domestic and international scholars. Enhancing the level of intellectual property protection can effectively raise a country's innovation level and improve the structure of intellectual property trade [3]. In terms of institutional distance, since first introduced the concept of institutional distance, numerous scholars have engaged in discussions on the relationship between institutional distance and international trade [4]; Based on the New-New Trade Theory and New Institutional Economics, overall institutional quality can positively impact international trade through means such as transaction costs and incomplete contracts [5]. From an economic institutional perspective, the average institutional distance constructed can affect foreign trade through the suppression of transaction costs [6]. From the viewpoint of New Institutional Economics, formal and informal institutional distances have different effects on service trade [7]; From the perspective of multinational enterprises, institutional distance significantly affects the performance of overseas subsidiaries [8]. In the context of bidirectional talent flow in technology between countries, institutional distance plays a significant role [9]. Different levels of institutional distance have varied impacts on economic activities; countries' outward investments exhibit characteristics of "political institutional proximity" and "economic institutional escape" [10]. Another portion of the literature provides evidence of the role of trade agreements in promoting intellectual property trade. Trade agreements that include levels of intellectual property protection can facilitate inter-country intellectual property trade and promote the upgrading of their industrial chains [11]; Regional trade agreements can enhance the facilitating role of institutional quality in intellectual property trade [12]; Regional trade agreements can help bridge institutional gaps and mitigate the impact of bilateral institutional distance [13]. For developing countries, the intellectual property clauses in trade agreements may lead to short-term losses due to high patent fees and lower returns [14]. There is also an analysis based on mega trade agreements such as RCEP, which offers recommendations for countries participating in the formulation of rules for intellectual property trade [15]. Regional trade agreements can enhance the facilitating role of institutional quality in intellectual property trade [12]. In terms of research methods, most scholars examined the probability of institutional distance and trade relationships based on traditional trade gravity models and spatial models, including discussions from various perspectives such as adjacent effects and national heterogeneity, the impact intensity of various institutions [16], controlling for culture and geographic distance [17]. However, as the intellectual property rights trade network is a highly important part of international trade networks, with the development of technology and information technology, traditional spatial and gravity models may find it difficult to detect the internal interdependencies and embedded external relationships. Overall, existing literature on the effects of institutional distance on various types of international economic activities is abundant, demonstrating the significant impact of institutional distance in international trade research. However, it still fails to validate the specific effects of institutional distance on intellectual property trade. Moreover, research on intellectual property trade networks is limited, and most existing literature relies on traditional trade gravity models, making it difficult to explore the network characteristics of intellectual property trade. Consequently, it is challenging to explain the forms of intellectual property trade between countries with different institutional distances in reality. Furthermore, in the international intellectual property trade landscape reshaped by multilateral trade agreements, the aforementioned studies still offer limited insights into understanding institutional distance, trade agreements, and intellectual property trade networks.

Therefore, this paper, based on the temporal exponential random graph model (TERGM), incorporates institutional distance and the intellectual property trade network into the research framework. Furthermore, it validates the model's conclusions using the generalized exponential random graph model (GERGM) to explore the mechanisms and characteristics of the dynamic development of the network. The innovation of this paper lies in the following aspects: (1) In terms of the research object, this paper differs from traditional studies of structural characteristics of bilateral trade relationships. Instead, it examines the formation of dyads and triads within the trade network as the object of study to explore the dynamic mechanisms of its evolution. This approach aims to deconstruct the embedded patterns of institutional distance matrix that influence the intellectual property trade network. (2) In terms of the research perspective, this paper takes a global perspective on the intellectual property trade network, controlling for potential influencing factors from both importing and exporting countries. It considers the roles of external scenarios and internal dependencies, resulting in more comprehensive and robust conclusions. (3) In terms of research methodology, this paper employs the exponential random graph model, which allows for the simultaneous control of various influencing factors and mechanisms, such as node attributes, internal, and external mechanisms. It also enables simulation based on model parameters, thereby providing a reference for the adjustment and improvement of the structure of the intellectual property trade network.(4) In terms of research framework, this article introduces a dynamic research framework that encompasses power relations, geopolitical strategies, and economic interests. It delves into the complexity inherent within institutional distances, intellectual property trade networks, and trade agreements from a deeper socio-political dimension.

···

Conclusion and Policy Recommendations

Based on data from the intellectual property trade network of 143 countries from 2008 to 2019, this paper employs a time-series Exponential Random Graph Model (TERGM) to examine the relationship between institutional distance and the intellectual property trade network. The research results indicate that a smaller institutional distance can reduce transaction costs and contract risks, making it easier for the two countries to engage in intellectual property trade. After controlling for factors such as geographical proximity, shared language, colonial relations, and self-organizing characteristics within the intellectual property trade network, the conclusion that reducing institutional distance promotes intellectual property trade still holds. The results of annual regression tests indicate that the impact of institutional distance on the intellectual property trade network fluctuates and increases over time. Research on the sub indicators of distance has found that the smaller the institutional distance in terms of political stability, government efficiency, and regulatory quality, the more likely it is to establish intellectual property trade relationships. Mechanism analysis shows that the institutional distance between the two countries will affect the establishment of intellectual property trade networks by influencing the signing of trade agreements between the two countries.

Based on the above conclusions, the recommendations are as follows: Firstly, actively reducing institutional distance with high-level institutional countries is conducive to developing intellectual property trade relations, according to the conclusions drawn in this paper. Therefore, under the premise of actively improving and developing the domestic institutional system, countries should align with the global development pattern, actively integrate into international economic and trade rules, continuously improve their intellectual property protection levels, promote domestic economic development and innovation, and gain long-term advantages in international resource allocation and participation in global division of labor. Secondly, attaching importance to intellectual property trade with countries at similar institutional levels within the region. Based on the research findings of this paper, the impact of institutional distance is expanding year by year. Therefore, countries should pay more attention to domestic institutional reforms. Combining the self-organizing characteristics of intellectual property trade networks, they should further focus on regional economic trade, promote the process of regional economic integration, and strive to expand the marginal of intellectual property trade networks. Efficiently utilize the international market space and trade cooperation relations with neighboring countries to reduce trade costs and improve market efficiency. Thirdly, actively promoting the signing and implementation of free trade agreements. According to the research findings of this paper, countries with smaller institutional distances are more likely to sign trade agreements, thereby promoting intellectual property trade between the two countries. It is conducive to the targeted promotion of the development of domestic institutional construction, facilitating the attraction of trade agreement relations with more countries, and maximizing the trade promotion effects brought about by such agreements. Fourthly, there is a possibility for countries to increase institutional distance in intellectual property trade by signing trade agreements. The conclusion of this article suggests that institutional distance may promote intellectual property trade through trade agreements. In the context of hindered globalization, bilateral or multilateral trade agreements can p

---

## [Decision Letter · Decision Letter 2]

5 Aug 2024

Institutional Distance, Trade Agreements, and Intellectual Property Trade Networks: Evidence from Cross-Border Data

PONE-D-23-37149R2

Dear Dr. Li,

We’re pleased to inform you that your manuscript has been judged scientifically suitable for publication and will be formally accepted for publication once it meets all outstanding technical requirements.

Kind regards,

Zahra Masood Bhutta

Academic Editor

PLOS ONE

Reviewers' comments:

Reviewer's Responses to Questions

**Comments to the Author**

1. If the authors have adequately addressed your comments raised in a previous round of review and you feel that this manuscript is now acceptable for publication, you may indicate that here to bypass the “Comments to the Author” section, enter your conflict of interest statement in the “Confidential to Editor” section, and submit your "Accept" recommendation.

Reviewer: All comments have been addressed

2. Is the manuscript technically sound, and do the data support the conclusions?

Reviewer: Yes

3. Has the statistical analysis been performed appropriately and rigorously? 

Reviewer: Yes

4. Have the authors made all data underlying the findings in their manuscript fully available?

Reviewer: Yes

5. Is the manuscript presented in an intelligible fashion and written in standard English?

Reviewer: Yes

6. Review Comments to the Author

Reviewer: (No Response)

7. PLOS authors have the option to publish the peer review history of their article (what does this mean?). If published, this will include your full peer review and any attached files.

---

## [Editor Report · Acceptance letter]

21 Aug 2024

PONE-D-23-37149R2 

PLOS ONE

Dear Dr. Li, 

I'm pleased to inform you that your manuscript has been deemed suitable for publication in PLOS ONE. Congratulations! Your manuscript is now being handed over to our production team.

Kind regards, 

on behalf of

Dr. Zahra Masood Bhutta 

Academic Editor

PLOS ONE